# Application of SWAT in Hydrological Simulation of Complex Mountainous River Basin (Part II: Climate Change Impact Assessment)

**Suresh Marahatta [1,\*], Deepak Aryal [1], Laxmi Prasad Devkota [2,3], Utsav Bhattarai [3,4] and Dibesh Shrestha [3]**

[1] Central Department of Hydrology and Meteorology, Tribhuvan University, Kathmandu 44600, Nepal; deepak.aryal@cdhm.tu.edu.np

[2] Nepal Academy of Science and Technology (NAST), Kathmandu 44600, Nepal; lpdevkota1@gmail.com

[3] Water Modeling Solutions Pvt. Ltd. (WMS), Kathmandu 44600, Nepal; utsav.bhattarai@wms.com.np (U.B.); dibeshshrestha@live.com (D.S.)

[4] Institute for Life Sciences and the Environment, University of Southern Queensland, Toowoomba, QLD 4350, Australia

\* Correspondence: suresh.marahatta@cdhm.tu.edu.np

**Abstract:** This study aims at analysing the impact of climate change (CC) on the river hydrology of a complex mountainous river basin—the Budhigandaki River Basin (BRB)—using the Soil and Water Assessment Tool (SWAT) hydrological model that was calibrated and validated in Part I of this research. A relatively new approach of selecting global climate models (GCMs) for each of the two selected RCPs, 4.5 (stabilization scenario) and 8.5 (high emission scenario), representing four extreme cases (warm-wet, cold-wet, warm-dry, and cold-dry conditions), was applied. Future climate data was bias corrected using a quantile mapping method. The bias-corrected GCM data were forced into the SWAT model one at a time to simulate the future flows of BRB for three 30-year time windows: Immediate Future (2021–2050), Mid Future (2046–2075), and Far Future (2070–2099). The projected flows were compared with the corresponding monthly, seasonal, annual, and fractional differences of extreme flows of the simulated baseline period (1983–2012). The results showed that future long-term average annual flows are expected to increase in all climatic conditions for both RCPs compared to the baseline. The range of predicted changes in future monthly, seasonal, and annual flows shows high uncertainty. The comparative frequency analysis of the annual one-day-maximum and -minimum flows shows increased high flows and decreased low flows in the future. These results imply the necessity for design modifications in hydraulic structures as well as the preference of storage over run-of-river water resources development projects in the study basin from the perspective of climate resilience.

**Keywords:** climate change; fractional difference; SWAT; quantile mapping; extreme flow

## 1. Introduction

Traditional energy sources along with human labour and draught transport were replaced initially by coal and then by oil in the early 1900s for powering machines and transportation [1,2]. Access to cheaper fossil fuels has been a major milestone for modern development pathways [3]. Since the beginning of the industrial age, the ability to harness and use different forms of energy has led to global economic growth and an increase in production and consumption, which enabled people to perform increasingly productive tasks and to improve the living standards of billions of people [4,5]. However, scientific evidence indicates that huge emissions of $CO_2$ and other greenhouse gases (GHGs) in the atmosphere are associated with the increasing use of fossil fuel [6,7]. The era of the industrial revolution can, thus, be taken as the starting point of climate change (CC) as the scientific community has defined it today [8]. With the widespread use of fossil fuels and

human interventions after the first industrial revolution, global warming has emerged as an environmental issue that has captured the attention of the world [9].

CC has become a major challenge and threat to the planet, particularly after World War II. Although alteration to the Earth's climate has been going on forever, we have started acknowledging its impacts on humans and the current environment only in the second half of the 20th century [9]. The first world conference on the environment held in Stockholm in June 1972 [10] formally opened the doors for a dialogue between industrialized and developing countries on the link between economic growth; the pollution of the air, water, and oceans; and the wellbeing of people around the world. Moreover, the formation of the United Nations Environment Programme (UNEP) is one of the major achievements of the Stockholm conference. The UN World Meteorological Organization (WMO) and UNEP established the Intergovernmental Panel on Climate Change (IPCC) with the objective of conducting and disseminating the findings of scientific research on CC in 1988 [11]. The Second Earth Summit organized by the UN in Rio de Janeiro, Brazil, in June 1992 formed a mechanism for cooperation between states, sectors, and people on issues related to environmental protection and sustainable economic development [12]. Further, in 1997, the Kyoto Protocol set the first GHG emission reduction targets for industrialized nations. In this regard, a total of 192 nations of the world committed to reducing their emissions by an average of 5.2% by 2012, which is popularly referred to as Kyoto Protocol [13]. The United Nations Climate Change Conference held in December 2009 documented that CC is one of the greatest challenges of the present day and prescribed that actions be taken to keep temperature increases to below 2 °C [14]. The 2015 Paris COP 21, a global consensus, established that average global warming needs to be kept below 1.5 to 2 °C to avoid the irreversible threat to environmental, economic, social, and political challenges by CC for years and decades to come [15]. The IPCC reports that all currently available global climate models (GCMs) agree on an increase in global mean temperature over the 21st century. The latest assessment report by the IPCC (AR6) has recommended limiting global warming to 1.5 °C in order to significantly reduce the risks and impacts of CC [16].

The GCMs and regional climate models (RCMs) have been found to be effective tools in developing a better understanding of CC by predicting future climate projections [17,18]. Scientists of the Geophysical Fluid Dynamics Laboratory (GFDL), National Oceanic and Atmospheric Administration (NOAA), developed the first coupled ocean-atmosphere general circulation climate model (GCM) in the 1960s capable of simulating the temperatures and precipitation of the past 50 years [19,20]. Following GFDL, many other studies across the world have developed different climate models. For example, the Hadley Centre Global Environment Model (HadGEM) [21], the Canadian Earth System Model (CanESM) [22], and the Max Planck Institute for Meteorology (MPI) [23] are some notable GCMs, while the Seoul National University Regional Climate Model (SNURCM) [24], the Max Planck Institute for Meteorology—REMO2009 [25], and the Conformal Cubic Atmospheric Model (CCAM) [26] are popular RCMs. Lately, it has become very convenient to use these GCM/RCM datasets in hydrological models such as the Système Hydrologique Européen (MIKE SHE), the Soil and Water Assessment Tool (SWAT), and the Variable Infiltration Capacity (VIC), among others, for CC studies. Comparing future climate projections with the baseline period to reach meaningful conclusions has been a routine procedure in the hydrological modelling and CC domains. Water availability studies using the output of climate models have been carried out at global [27], regional [28], and local scales [29]. Several studies have been conducted to assess the water availability and impacts of CC in the Hindu Kush Himalayan region [30–33] that includes the Budhigandaki River Basin (BRB) [34–36]. Results of such studies vary considerably across the spatial and temporal scales, and thus a generic conclusion on the impact of CC in water availability cannot be reached deterministically. These studies suggest the SWAT model can be a useful tool to assess the flow and water balance and the impacts of CC on them.

The quantification of available water at the local scale and examining how it is impacted by CC is extremely important from a water management perspective at the river

basin level. The overarching objective of this study is to analyse the impact of CC on the hydrology of a complex mountainous river basin, namely the Budhigandaki River Basin (BRB) of Nepal. The well-calibrated and validated SWAT model (in Part I of this research) was used to simulate future flows using future CC data [37]. A relatively new approach of selecting climate models representing four different extreme cases (warm-wet, cold-wet, warm-dry, and cold-dry conditions) were applied in the study. Generally, analysis of the impact of CC was limited to annual, monthly, and seasonal flows and based on people's perceptions in most of the previous studies [32,34,38–40]. This study has extended the analyses further to compare with the baseline condition the results of the frequency analysis of the annual one-day-maximum and -minimum flows for three-time windows, i.e., immediate, mid, and far futures.

## 2. Methodology and Data

The methodological framework used in this study is depicted in Figure 1. Future data of different global climate models (GCMs) to assess the impact of climate change (CC) on the Budhigandaki River Basin (BRB) was projected for 79 years (2021–2099). These climate data were considered for two RCPs, 4.5 (stabilization scenario) and 8.5 (high emission scenario), of the Intergovernmental Panel on Climate Change Fifth Assessment Report (IPCC AR5) [41]. A brief description of the selected GCMs and their selection criteria, bias correction methods, impact of CC on climatic parameters and hydrology, and frequency analysis of extreme flows are given below.

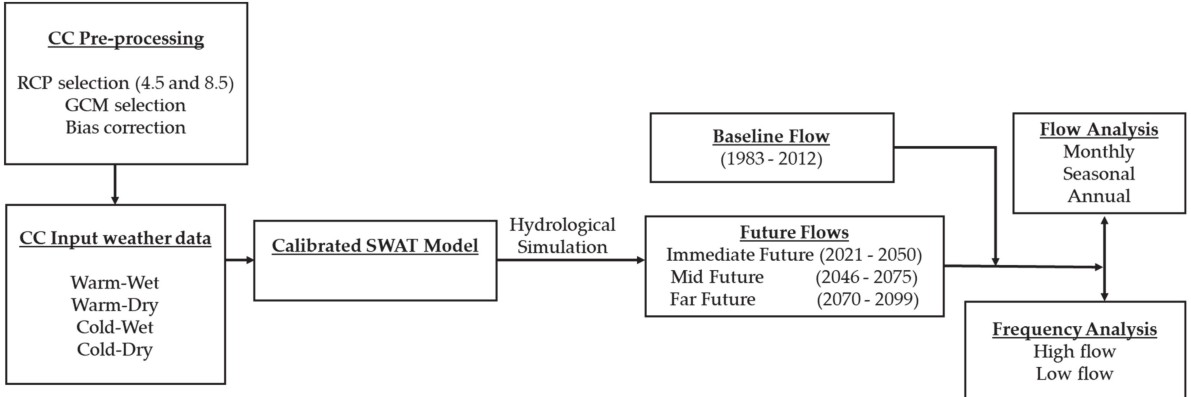

**Figure 1.** Overall methodological framework.

### 2.1. Hydrological Modelling

The Soil and Water Assessment Tool (SWAT), a continuous-time, semi-distributed, process-based river basin simulation model [42,43] capable of simulating hydrology and other environmental processes, was used in the study to examine the impact of CC on the basin hydrology. The Budhigandaki River Basin (BRB) was divided into 16 sub-basins and 344 hydrologic response units (HRUs) to capture the spatial heterogeneity across the basin. The model was calibrated and validated at the Arughat hydrological station. Moreover, supplementary validation of the model was done at three locations upstream and downstream of Arughat. The model performance was evaluated using four widely used statistical indicators: Nash–Sutcliffe efficiency (NSE), root mean square standard deviation ration (RSR), percent bias (PBIAS), and Kling–Gupta efficiency (KGE). Details of the SWAT hydrological model development and its evaluation are discussed in the first part of this study [37].

### 2.2. Selection of Climate Models

This study used the advanced envelop-based climate selection method to assess the projected future climates as described in Lutz et al. [28]. It is based on two general criteria for the selection of GCMs: (a) GCMs should be common to a pool of models with changes

in temperature and precipitation, and (b) they must be available on a continuous daily scale. Based on these two conditions, 105 models for RCP 4.5 and 78 models for RCP 8.5 scenarios were included in the initial pool of models. Climatic data (mean temperature and precipitation) were obtained from GCMs through the KNMI climate explorer interface (https://climexp.knmi.nl/start.cgi, accessed on 16 June 2020) for the study area. The applied method uses three steps connecting future projections with past performance (1981–2005), as shown in Figure 2.

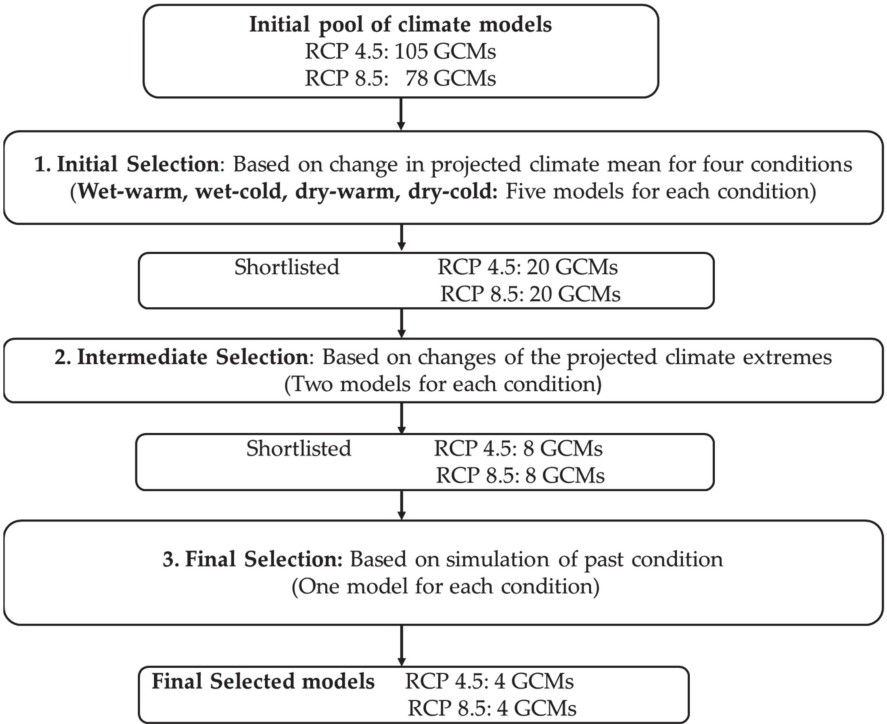

**Figure 2.** Climate model selection (adopted and modified from [28]).

**Step 1**: In the first step, the projected changes in future climate from the baseline period are computed for each GCM listed in the pool of models considering warm-dry, warm-wet, cold-wet, and cold-dry corners to form an envelope. The 10th and 90th percentile values for change in mean temperature ($\Delta T$) and percentage change in annual precipitation ($\Delta P$) for each scenario were determined. These values represent the four corners of the spectrum of the projections for temperature and precipitation change. The tenth percentile value of $\Delta T$ and tenth percentile value for $\Delta P$ are in the 'cold-dry' corner of the spectrum. Likewise, the tenth percentile value of $\Delta T$ and 90th percentile value for $\Delta P$ are in the 'cold-wet' corner, while the 90th percentile value of $\Delta T$ and 90th percentile value of $\Delta P$ are in the 'warm-wet' corner. Similarly, the 90th percentile value of $\Delta T$ and tenth percentile value of $\Delta P$ fall in the 'warm-dry' corner of the spectrum. The proximity of the model running to the 10th and 90th percentile values is derived using the distance metrics recommended by [28]. A total of 20 models (5 models in each four corners) were selected from this step.

**Step 2**: Two GCMs in each corner were selected in the intermediate step based on the projection of changes in climate extremes using four of the Expert Team on Climate Change Detection and Indices (ETCCDI), as mentioned by [28]. They are $R_{95P}$ [Precipitation due to extremely wet days (> 95th percentile)], CDD [Consecutive dry days: maximum length of dry spell ($p < 1$ mm)], WSDI [Warm spell duration index: count of days in a span of at least 6 days where TX > 90th percentile ($TX_{ij}$ is the daily $T_{max}$ on day i in period j)], and CSDI [Cold spell duration index: count of days in a span of at least 6 days where TN < 10th percentile ($TN_{ij}$ is the daily $T_{min}$ on day i in period j)].

**Step 3**: In the final step, the ability of the selected GCMs to hindcast the past observation is evaluated between the selected two models in Step 2 of each of the four corners. In this step, the Taylor method [44] was used to evaluate the GCMs on the basis of three statistics: correlation coefficient, centred root mean squared error (RMS), and the standard deviation. The most reasonable GCM for each corner was selected as the one having the average of the highest Taylor score of the precipitation and temperature. In this way, four models, one in each corner with the highest Taylor score, were selected for each RCP.

*2.3. Bias Correction*

In order to apply GCM projections at the local scale, bias correction is necessary. This is because of the following reasons: (a) GCMs have inherent systematic biases which can be due to the model's representation of physical processes and their parameterization, initializations, or human judgement, and (b) they are often incompatible on scales (because of coarser resolution) that are necessary for local level hydrological impact studies [45]. Different methods for bias correction for climate variables like precipitation and temperature are discussed in the literature that range from simple corrections in the annual or monthly mean values to complex distribution fitting that corrects the entire distribution. For example, [46] used the delta change approach in crop modelling in Europe for both temperature and precipitation. Lenderink et al. [47] used the linear scaling approach for estimating the future discharges of the Rhine River. Local intensity scaling (LOCI) was applied by [48], while the power transformation method was used in other studies [47,49].

This study used the distribution mapping approach in which the mean, variance, and whole distribution is considered [50–52]. The projected climate data at the meteorological stations were then bias corrected using the quantile mapping (QM) method [51,53]. QM corrects the quantiles of GCM data to match those of observed data by creating suitable transfer functions explained in Equation (1):

$$X^{corr}_{future,t} = inverse\ ecdf\ ^{obs}_{baseline}\left(ecdf^{GCM}_{baseline}\left(X^{GCM}_{future,t}\right)\right) \tag{1}$$

where, *ecdf* is the empirical cumulative distribution function for the reference time period, $X^{GCM}_{future,t}$ is the raw GCM (projected value) at future time t, $ecdf^{GCM}_{baseline}$ is the empirical cumulative distribution function of GCM for the baseline period, and *inverse ecdf* $^{obs}_{baseline}$ is the inverse empirical cumulative distribution function of observation for the baseline period. $X^{corr}_{future,t}$ is the corrected estimate of $X^{GCM}_{future,t}$. We used the frequency adaptation method as described in [54] for the correction of extra dry days when the frequency of dry days in the baseline period in GCM data is greater than the frequency of dry days in the observed data.

The bias-corrected times series climatic data from the selected GCMs for each meteorological station were averaged for each climate scenario. The projected future CC and associated impacts were analysed based on those individual climatic scenarios which were used as input to the SWAT hydrological model.

*2.4. Climatology under Climate Change*

The percentage change in annual average precipitation and mean temperature under CC were compared with the baseline data for both emission scenarios (RCPs 4.5 and 8.5).

2.4.1. Climate Change Impact Analysis of Future Flows

The SWAT model developed and discussed in Part I of this study [37] was applied to simulate the future flows by enforcing the projected GCMs'- CC data. Projected flows were divided into three 30-year time windows: Immediate Future (IF: 2021–2050), Mid Future (MF: 2046–2075), and Far Future (FF: 2070–2099). Projected flows were compared with corresponding monthly, seasonal, annual, and fractional differences of extreme flows ($Q_{90}$ and $Q_{10}$) of the simulated baseline flows (1983–2012). Such comparisons were made for all four scenarios, three future time windows, and for both RCPs.

### 2.4.2. Frequency Analysis

Annual one-day-maximum and -minimum flow series were extracted for each time window of all four scenarios for both RCPs along with the baseline simulated flow. Gumbel distribution [55] was fitted to these time series in order to find the magnitude of high and low flows for the selected return periods. Interestingly, some flood-related studies have applied hydro-economic [56] and techno-social [57] perspective CC assessment methods.

## 3. Results

### 3.1. Climate Model Selection

The first selection was made by comparing the projected mean temperature and annual precipitation changes (%) of 2021–2050 with the baseline period of 1981–2005 for both RCPs shown as dots in Figure 3. In RCP 4.5, the projected changes in annual precipitation are in the range of −9% to +23%, while most models show an increase in precipitation. Similarly, the change in projected temperature ranges from +0.6 °C to +3.1 °C with the multi-model mean showing an increase by approximately +1.7 °C. Similarly, in RCP 8.5, most of the models indicate an increase in annual precipitation, while in some cases, it is decreases (range of change in precipitation: −11% to +21%). In the case of temperature, the changes are from +0.7 °C to +3.0 °C, with the mean value of +1.9 °C.

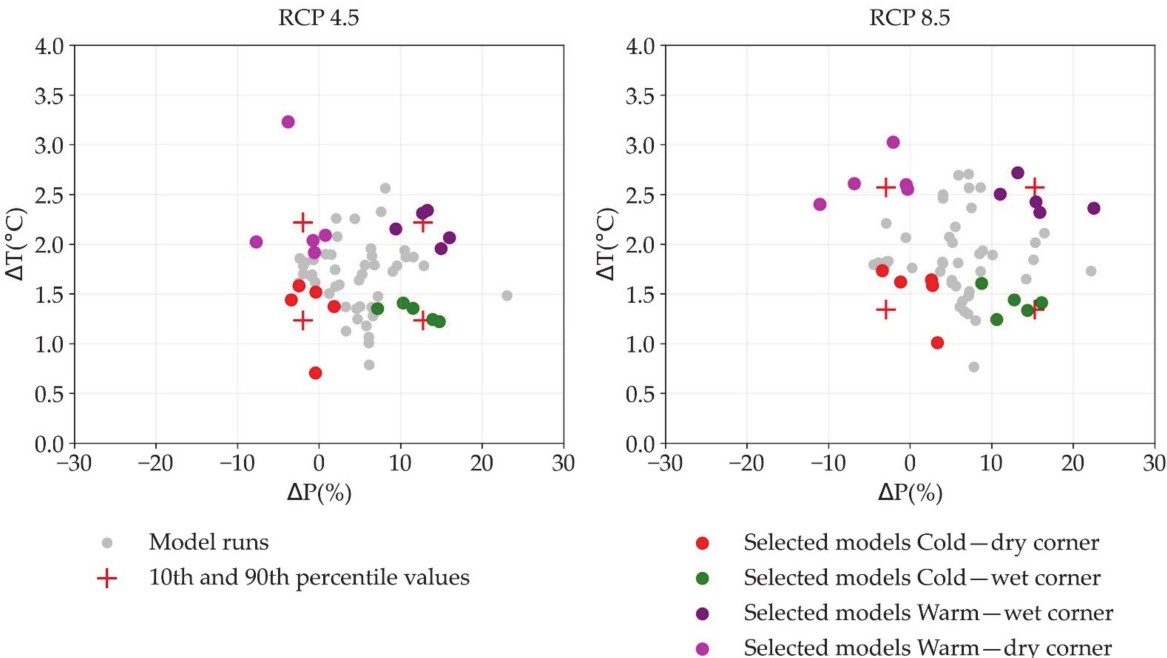

**Figure 3.** Projected changes in annual precipitation (ΔP %) and annual mean temperature (ΔT °C) for RCP 4.5 and RCP 8.5. Pink, violet, red, and green dots represent warm-dry, warm-wet, cold-dry, and cold-wet conditions, respectively.

ETCCDI extreme indices, viz. $R_{95P}$, CDD, WSDI, and CSDI, were used to filter GCMs from 20 to 8. The two GCMs that have the topmost combined scores for the changes in precipitation and temperature indices in each corner are selected in the intermediate selection steps given in Supplementary Tables S1 and S2. Using the Taylor method [44], four GCMs, one for each corner, are selected (Table 1). HadGEM2 was selected for cold and dry conditions, GFDL for cold and wet conditions, and CanESM2 for warm and wet conditions for both RCPs. However, in the case of warm and dry conditions, MPI-ESM for RCP 4.5 and MIROC-ESM for RCP 8.5 were selected.

### 3.2. Bias Correction

Monthly average precipitation and temperature (observed, uncorrected, and bias corrected) are given in Figure 4 for RCP 4.5 of CanESM2 as a representative graph. The graph shows that the climate data are well bias corrected.

**Table 1.** Summary of selected GCMs for RCPs 4.5 and 8.5.

| Climatic Condition | GCMs for RCP 4.5 | GCMs for RCP 8.5 |
|---|---|---|
| Cold-dry ($p_{10\_10}$) | HadGEM2-CC_rcp45_r1i1p1 | HadGEM2-ES_rcp85_r1i1p1 |
| Cold-wet ($p_{10\_90}$) | GFDL-ESM2G_rcp45_r1i1p1 | GFDL-ESM2M_rcp85_r1i1p1 |
| Warm-wet ($p_{90\_90}$) | CanESM2_rcp45_r3i1p1 | CanESM2_rcp85_r3i1p1 |
| Warm-dry ($p_{90\_10}$) | MPI-ESM-LR_rcp45_r3i1p1 | MIROC-ESM-CHEM_rcp85_r1i1p1 |

Note: $p_{x\_y}$: xth and yth percentiles of temperature and precipitation.

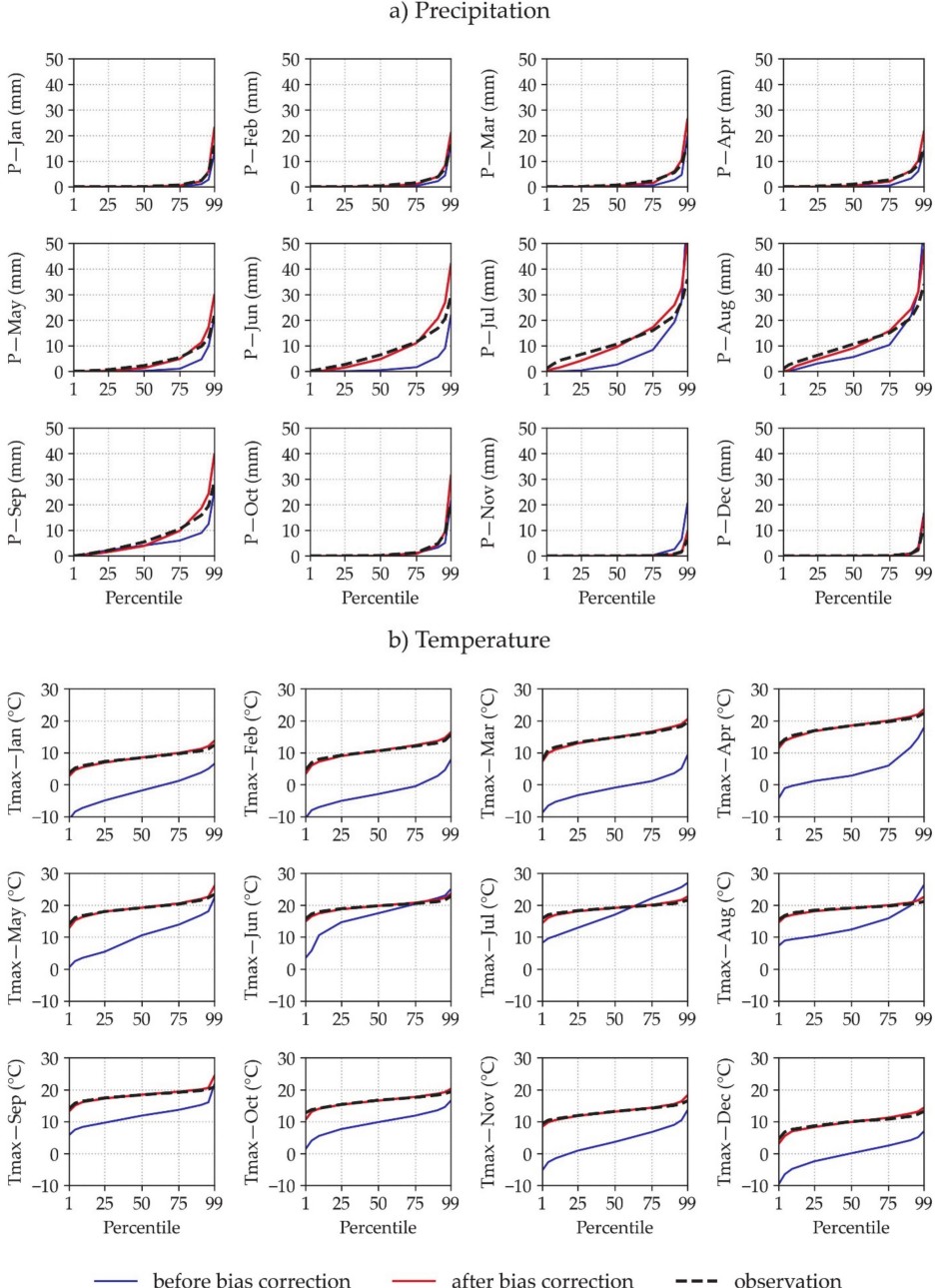

**Figure 4.** Observed (black), before bias correction (blue), and after bias correction (red) precipitation (top) and maximum temperature (bottom) monthly values for RCP 4.5 of CanESM2.

### 3.3. Climatology under Climate Change

The percentage changes in annual precipitation and temperature under the two emission scenarios (RCPs 4.5 and 8.5) with respect to the baseline values are depicted in Figure 5. The projected precipitation by all GCMs for all time windows (IF, MF and FF) is most likely to increase in both RCPs except MPI-ESM (MF, FF) and HadGEM2 (IF), in the case of RCP 4.5, and HadGEM2 (IF and MF) for RCP 8.5. The highest increase of about 9% and 20% from the baseline values are found for RCP 4.5 (in IF) and RCP 8.5 (in FF), respectively, projected by the CanESM2 model. The lowest projected precipitation by the HadGEM2 is found in the IF time window. Their values are about 5% and 7% lower than the base case, respectively, in RCPs 4.5 and 8.5. It can be seen from the figure that the precipitation is expected to increase with time (IF < MF < FF) in RCP 8.5. However, in the case of RCP 4.5, the response is mixed, i.e., an increasing trend for HadGEM2 (Cold-dry) and GFDL_ESM2G (Cold-wet) and a decreasing one in the other two cases (Warm-wet and Warm-dry). This characteristic is distinctly depicted in Figure 6, i.e., the projected long-term precipitation in RCP 8.5 shows a clear increasing trend which is not clear for RCP 4.5.

Both the maximum and minimum temperature projections by all the selected GCMs (in all climatic conditions) are found increasing while moving from IF to FF in both emission scenarios. The rate of increase of temperature is higher by all projections for RCP 8.5 than that of RCP 4.5, as expected (Figures 5 and 6). It can also be observed that the increase in minimum temperature is more than that of maximum temperature in all time windows for both emission scenarios and in all conditions except the cold-wet condition (GFDL-ESM2G of RCP 4.5). However, the projected temperatures of selected models are found to be different. The maximum increase is found for the minimum temperature projected by MIROC-ESM-CHEM (warm-dry/FF) of RCP 8.5, i.e., 6.5 °C. The minimum increase is found to be 0.6 °C projected by the GFDL-ESM2G model in IF (Figure 5).

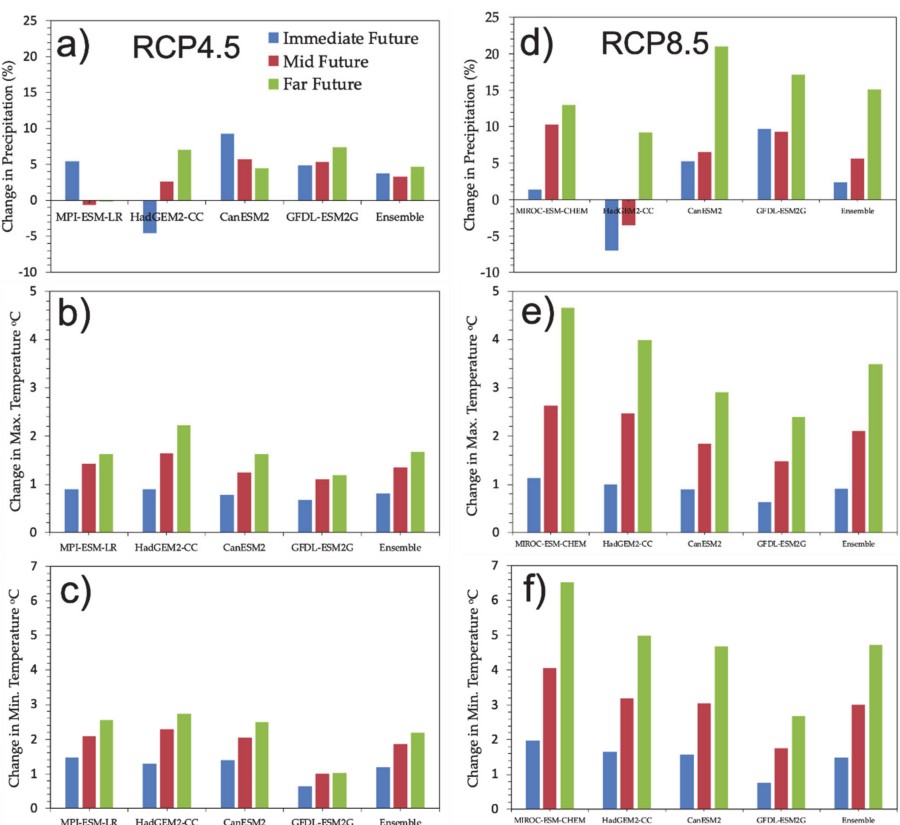

**Figure 5.** Change in precipitation (**a**,**d**) and change in temperature [maximum (**b**,**e**), minimum (**c**,**f**)] pattern of the four selected GCMs and ensemble of RCPs 4.5 (left) and 8.5 (right) with their respective time windows: blue (IF), red (MF), and green (FF).

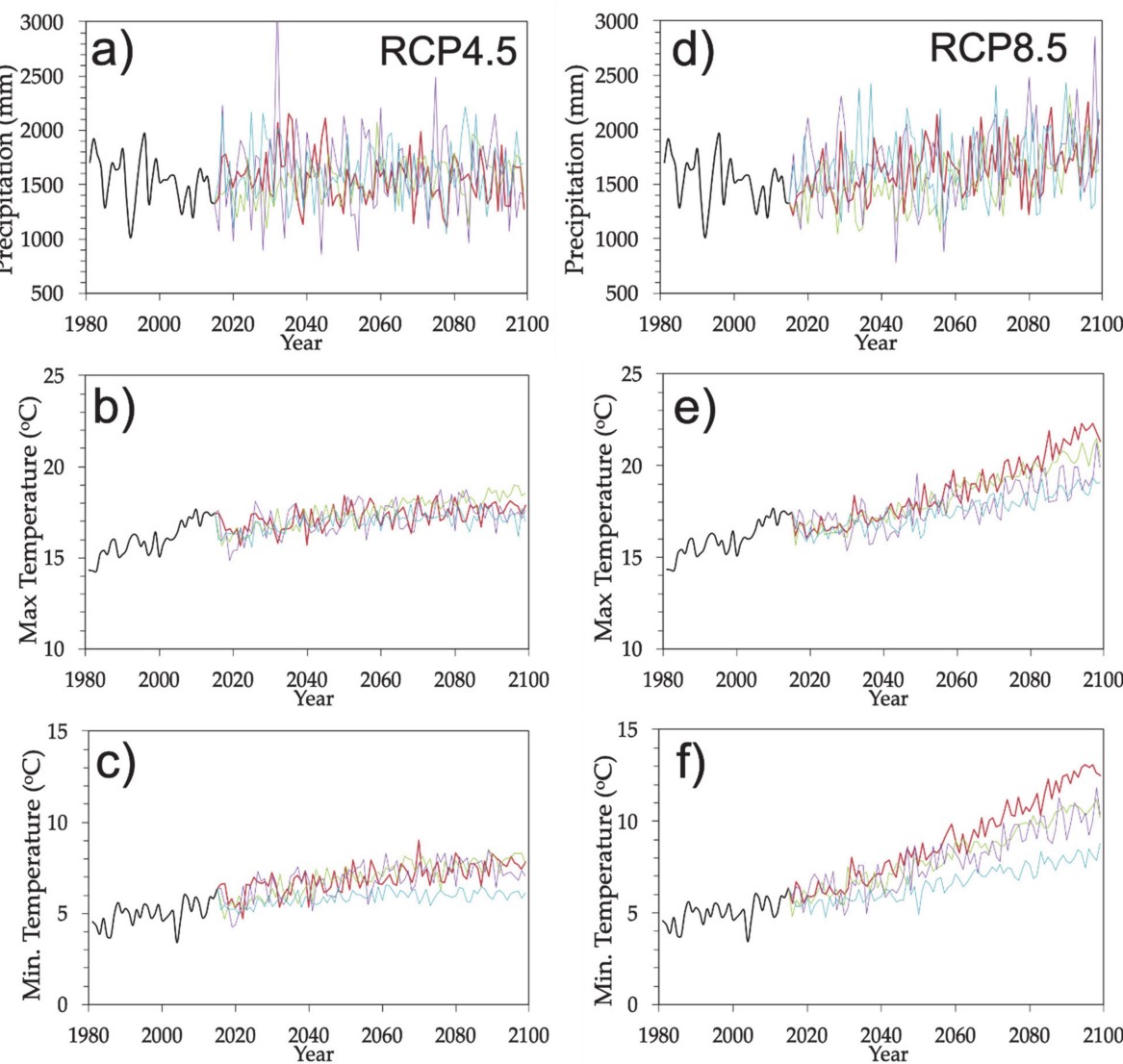

**Figure 6.** Precipitation and temperature (observed, bias corrected) of RCP 4.5 (**a–c**) and 8.5 (**d–f**) of the selected GCMs; black (baseline), red (MIROC/MPI), blue (GFDL), green (HadGEM), and purple (CanESM2).

### 3.4. General Hydrology under Climate Change

The simulated average annual flow at the outlet of BRB for the baseline period is 240 m$^3$/s [37]. Table 2 presents the predicted flow and corresponding percentage changes by all selected GCMs for four climatic conditions (warm-wet: projected by CanESM2; warm-dry: MPI (RCP 4.5/MIRCO (RCP 8.5), cold-wet: GFDL, and cold-dry: by HadGEM2) and their ensemble for all time windows.

The maximum increase of annual flow in RCP 4.5 is more than 30% in the cold-dry condition of FF and the minimum increase is about 10% in IF for the same condition predicted by the HadGEM2 model. It shows that the HadGEM2 predicted flow has a higher variability range than other GCMs in terms of annual averages. The increased flow of the long-term annual average is almost the same for the warm-wet condition projected by CanESM2 (IF: 29%, MF: 30% and FF 28%), whereas it has a decreasing trend for the warm-dry condition projected by the MPI model (IF: 26%, MF: 20% and FF: 17%). The flow of the other remaining two conditions have an increasing trend while moving from IF to FF [GFDL: 18% (IF), 20% (MF) and 24% (FF); and HadGEM2: 10% (IF), 24% (MF) and 31% (FF)].

**Table 2.** Impact of climate change on long-term annual flow.

| Conditions | Time Window | RCP 4.5 | | RCP 8.5 | |
|---|---|---|---|---|---|
| | | Flow (m³/s) | % Change | Flow (m³/s) | % Change |
| Baseline | | 240 | - | 240 | - |
| Cold-Wet (GFDL-ESM2G) | Immediate Future | 283 | 18 | 304 | 27 |
| | Mid Future | 287 | 20 | 317 | 33 |
| | Far Future | 297 | 24 | 358 | 49 |
| Warm-Wet (CanESM2) | Immediate Future | 309 | 29 | 311 | 30 |
| | Mid Future | 311 | 30 | 315 | 32 |
| | Far Future | 306 | 28 | 377 | 57 |
| Cold-Dry (HadGEM) | Immediate Future | 263 | 10 | 251 | 5 |
| | Mid Future | 297 | 24 | 272 | 14 |
| | Far Future | 314 | 31 | 331 | 38 |
| Warm-Dry (MPI-ESM-LR/MIROC-ESM) | Immediate Future | 301 | 26 | 287 | 20 |
| | Mid Future | 288 | 20 | 334 | 39 |
| | Far Future | 281 | 17 | 350 | 46 |
| Ensemble | Immediate Future | 289 | 21 | 288 | 20 |
| | Mid Future | 296 | 23 | 310 | 29 |
| | Far Future | 299 | 25 | 354 | 48 |

The long-term average annual flow predicted by all GCMs for all time windows in RCP 8.5 are also more than the baseline flow, in increasing order from IF to FF for all climatic conditions, similar to RCP 4.5. The increase in annual projected flow is between 5% (cold-dry/IF) and 57% (warm-wet/FF).

Thus, it can be observed that the long-term average annual flows are projected to increase in all climatic conditions for both RCPs. The range of increment of ensembled flow is in between 21% and 25% in RCP 4.5 and 20% and 48% in RCP 8.5. This shows that the magnitude of increment of future flow is expected to be more for the higher emission scenario.

*3.5. Variation in Monthly Flows*

Knowledge on the monthly variability under CC is useful for risk assessment of water resource development projects, such as hydropower, irrigation, and municipal water supplies. The long-term monthly flow for the three-time windows projected by the four GCMs representing four climatic conditions of RCP 4.5 and RCP 8.5 emission scenarios are given in Supplementary Tables S3 and S4, respectively. The data show that the range of change in monthly projected flow in RCP 4.5 is from −17% (June/MF/cold-dry) to 68% (March/FF/warm-wet) with respect to the baseline flows. It is noted here that only 4 out of 144 cases (12 months × time windows × 4 climatic conditions) have more than 5% decrease in monthly flows. In almost 60% of the cases, the flows are in the range of +10% to +25% of the baseline flows. Monthly flows more than 25% of the baseline flows are found to be in one fourth of the total number of months in the simulated period. In general, in three climatic conditions (warm-wet, cold-wet, and warm-dry), more than 25% increase is found in the month of March. Even for the cold-dry condition, the increased percentage is high (>15%). The monsoon flow is expected to increase significantly in all time windows except for the cold-dry condition in June. The highest absolute magnitude of the projected monthly maximum flow in all time windows for all climatic conditions is found to occur in August, similar to the baseline, except for the far-future of the cold-wet condition. In this case, the flow value is at its maximum in July.

In the case of RCP 8.5, the range of change of the monthly flow is from nearly −50% (July/MF/cold-dry) to 200% (October and November/MF/cold-dry). Almost two-thirds

of the predicted monthly flows are more than 25% of the baseline values for all climatic conditions. Almost one-fourth of the cases are between a 10 and 25% increase in monthly flows. The decrease in predicted flow was found only for cold-dry conditions. It is interesting to note that this decrease is mainly observed in the monsoon season (June, July, and August) and in May. The rate of increase is found to be higher in the post-monsoon season (October and November).

The monthly baseline and predicted ensembled flows as well as the corresponding percentage changes due to CC for both RCPs are given in Figure 7.

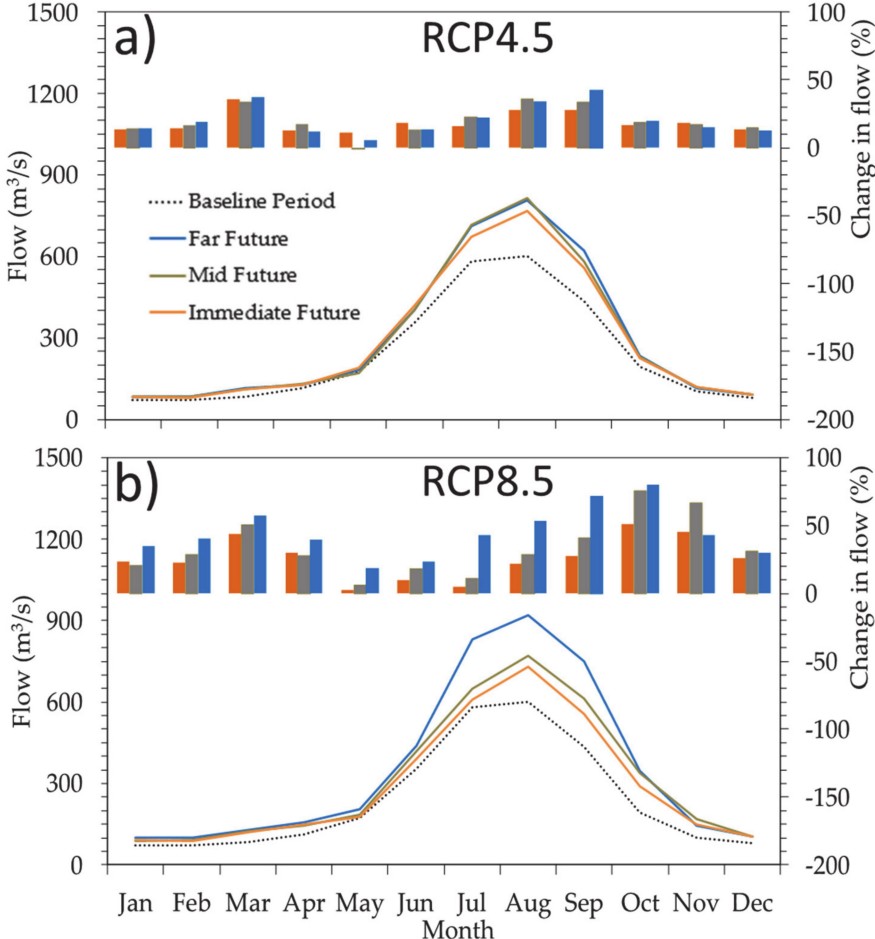

**Figure 7.** Hydrographs of monthly ensembled and baseline flows and their changes; (**a**: RCP 4.5 and **b**: RCP 8.5) black dashed (baseline), red (Immediate Future—IF), grey (Mid Future—MF), and blue (Far Future—FF).

The ensembles of the projected mean monthly flow are expected to increase in March, August, September, October, and November for all scenarios except MF (May) of RCP 4.5 (2021–2099). On the other hand, in RCP 8.5, all the ensembles of the projected monthly flow increase in all time windows. The relative changes in the projected mean monthly flow under RCP4.5 are +11 to +28%, −1 to +36%, and +5 to +43% for IF, MF, and FF, respectively. These figures range from +3 to +52%, + 6 to + 77%, and +18 to + 82% for IF, MF, and FF, respectively, in the case of RCP 8.5. Ensembled flows are found to be higher in RCP 8.5 than in RCP 4.5, except for a few months (May to August of IF and July and August of MF). The maximum increases of 43% (September) in RCP 4.5 and 82% in RCP 8.5 occur towards the end of the century.

### 3.6. Variation in High and Low Flows

The 10th percentile ($Q_{10}$, high flow) and 90th percentile ($Q_{90}$, low flow) values were derived from the corresponding baseline and flow duration curves predicted by all four GCMs for the considered time windows. Their fractional differences ($Q_{10}$:$Q_{90}$) were calculated separately for each case. The 10th and 90th percentiles of baseline flow are, respectively, 598 m$^3$/s and 64 m$^3$/s. This means that the fractional difference of the baseline flow is 9. For RCP 4.5, the fractional differences of the ensembled flow are found to be 11, 12, and 12 for IF, MF, and FF, respectively. However, such ensemble values for RCP 8.5 are, respectively, 14, 17, and 18 for IF, MF, and FF. This result shows that variability is expected to increase with time and be of a higher magnitude in RCP 8.5 than in RCP 4.5.

The number of days exceeding $Q_{10}$ flow and less than $Q_{90}$ were calculated from the results to assess the frequency of incidence of high flow and low flow. The number of days exceeding $Q_{10}$ or not exceeding $Q_{90}$ was 1098 for the baseline period. The number of days (for all the four GCMs) that are expected to have flow exceeding $Q_{10}$ are more than 1098 days and not exceeding $Q_{90}$ are less than 1098 days, except for one case (FF-low flow-warm dry-RCP 4.5). The increase in the number of such days for IF, MF, and FF are, respectively, 58%, 66%, and 71% for RCP 4.5 and 43%, 62%, and 97% for RCP 8.5. On the other hand, the percentage of decrease in number of days in which the flow is expected to be less than $Q_{90}$ are 29%, 26%, and 14% for RCP 4.5 and 43%, 20%, and 42% for RCP 8.5 for IF, MF, and FF, respectively. This result indicates that the likelihood of the number of flooding days would increase during the high flow season, while the number of firm flow days in the low flow season would decrease, both instances pointing towards the negative impact of CC.

### 3.7. Frequency Analysis of Flow

In this study, the frequency analysis of annual one-day-maximum and -minimum flows at the outlet of the BRB was carried out by the Gumbel Method for the baseline and CC cases as discussed below.

#### 3.7.1. One-Day-Maximum Flow

Maximum instantaneous flows are generally used to estimate design floods [58,59]. Annual one-day-maximum floods and instantaneous floods are positively correlated [59–61]. Therefore, it is assumed that the impact of CC on instantaneous flows is the same as that in the annual one-day-maximum flow.

Changes in the one-day-maximum flood for the different conditions and return periods are given in Table 3, including baseline values. The baseline floods for the 100-, 500-, and 1000-year return period are, respectively, 1544, 1801, and 1911 m$^3$/s. Table 3 shows that the one-day-maximum flood resulting from CC is higher than the baseline floods for all climatic conditions and in all time windows for the considered return periods. However, the magnitude of predicted values are different depending on the selected GCMs. Such an increase is found in the range of 66% (warm-dry/IF/100 years) to 226% (warm-wet/MF/1000 years) for RCP 4.5 and 69% (cold-dry/IF/100 years) to 317% (cold-wet/FF/1000 years) in the case of RCP 8.5. It is noted here that flood magnitudes are more in RCP 8.5 than RCP 4.5, except in the IF of the cold-dry condition. On an average, the change in ensembled projected flood is around 110%, 125%, and 140% of baseline floods, respectively, for IF, MF, and FF time windows in the case of RCP 4.5. These changes are about 150%, 200%, and 250%, respectively for IF, MF, and FF time windows in RCP 8.5.

#### 3.7.2. One-Day-Minimum Flow

One-day-minimum simulated flows for the different scenarios are given in Table 4. Results showed that the range of change in one-day-minimum flow due to CC with respect to the baseline condition is between −27% (warm-dry/FF/20 years) and +9% (warm-wet/IF/2 years) for RCP 4.5, while it is between −20% (warm-wet/MF/20 years) and +16% warm-wet/FF/2 years) for the RCP 8.5 scenario. Almost half of the flow values in

RCP 4.5 are expected to decrease by more than 10%. This phenomenon is mainly observed in MF and FF. Similar results are obtained in RCP 8.5, i.e., almost 50% of the low flows are lower than the base case values. It is more prominent in MF. Such a decrease is mainly clustered for the 10- and 20-year return period flows. The maximum decrease in predicted ensembled one-day-low flows is observed in FF in the case of RCP 4.5 and MF in the case of RCP 8.5.

**Table 3.** One-day-maximum flood frequency analysis.

| Time Window | Return Period (Years) | Baseline Flow (m³/s) | % Change in Flow (RCP 4.5) | | | | | % Change in Flow (RCP 8.5) | | | | |
|---|---|---|---|---|---|---|---|---|---|---|---|---|
| | | | Warm and Dry | Cold and Dry | Warm and Wet | Cold and Wet | Ensembled | Warm and Dry | Cold and Dry | Warm and Wet | Cold and Wet | Ensembled |
| IF | 100 | 1544 | 66 | 101 | 153 | 102 | 106 | 204 | 69 | 159 | 137 | 142 |
| | 500 | 1801 | 67 | 109 | 163 | 106 | 111 | 226 | 72 | 171 | 145 | 154 |
| | 1000 | 1911 | 68 | 111 | 166 | 108 | 113 | 234 | 73 | 175 | 148 | 158 |
| MF | 100 | 1544 | 71 | 126 | 205 | 79 | 120 | 215 | 167 | 180 | 200 | 190 |
| | 500 | 1801 | 72 | 131 | 220 | 79 | 125 | 228 | 183 | 189 | 217 | 204 |
| | 1000 | 1911 | 72 | 133 | 226 | 78 | 127 | 233 | 188 | 192 | 223 | 209 |
| FF | 100 | 1544 | 114 | 141 | 183 | 97 | 134 | 238 | 175 | 269 | 285 | 242 |
| | 500 | 1801 | 123 | 149 | 197 | 98 | 142 | 254 | 183 | 280 | 309 | 256 |
| | 1000 | 1911 | 126 | 151 | 202 | 98 | 144 | 260 | 186 | 283 | 317 | 262 |

**Table 4.** One-day-minimum flow frequency analysis.

| Time window | Return Period (Years) | Baseline Flow (m³/s) | % Change in Flow (RCP 4.5) | | | | | % Change in Flow (RCP 8.5) | | | | |
|---|---|---|---|---|---|---|---|---|---|---|---|---|
| | | | Warm and Dry | Cold and Dry | Warm and Wet | Cold and Wet | Ensembled | Warm and Dry | Cold and Dry | Warm and Wet | Cold and Wet | Ensembled |
| IF | 2 | 56 | 7 | 0 | 9 | 4 | 5 | 5 | 2 | 9 | 7 | 6 |
| | 10 | 47 | 2 | 0 | −9 | −4 | −3 | 6 | −2 | −4 | 4 | 1 |
| | 20 | 45 | 0 | 0 | −13 | −7 | −5 | 7 | −2 | −7 | 4 | 1 |
| MF | 2 | 56 | −4 | 5 | 4 | −7 | 0 | 0 | −2 | 4 | −2 | 0 |
| | 10 | 47 | −11 | 0 | −11 | −15 | −9 | −11 | −15 | −15 | −15 | −14 |
| | 20 | 45 | −13 | −2 | −16 | −16 | −12 | −13 | −20 | −20 | −18 | −18 |
| FF | 2 | 56 | −13 | 2 | −2 | −5 | −4 | −5 | 4 | 16 | 4 | 4 |
| | 10 | 47 | −23 | −4 | −17 | −15 | −15 | −15 | −13 | 2 | −2 | −7 |
| | 20 | 45 | −27 | −7 | −20 | −18 | −18 | −18 | −18 | −2 | −4 | −11 |

## 4. Discussion

Precipitation and temperature are considered to be the most important climatic variables influencing the water availability of a basin. The influence of CC has significant implications on water resource planning and management. Recent studies have used varying models and datasets (GCMs and/or RCMS) and climate scenarios to assess CC through precipitation and temperature in various parts of Nepal, considering different time windows to the end of this century [31,62–65]. Expected changes in precipitation are not consistent across the country; however, all these studies predicted an increase in precipitation with time. Our results are comparable with these, showing that projected precipitation by almost all GCMs for all time windows (IF, MF and FF) is most likely to increase in both RCPs. In the case of temperature, expected changes quantified by the aforementioned studies are not uniform in this region. Nevertheless, most have predicted a rise in both maximum and minimum temperature [29,39,64,65]. In this study too, both the maximum and minimum temperatures projection by all the selected GCMs (in all climatic conditions) are expected to increase while moving from IF to FF in both emission scenarios.

This study found that the projected annual flows by all GCMs and for all time windows are greater than the base case. A similar trend has been reported in other studies made in Nepalese rivers basins: for example, Indrawati [31], in Bagmati [62], Kaligandaki [66], Bheri [67], Karnali [29], and Koshi [32,40,59], except [39] in Tamor. Among these results, the trend of increase in ensembled future flows is surprisingly found similar to that of [40]. The percentage change in flow due to CC in IF/MF/FF of this study and [40] are, respectively, 21/23/25 and 16/22/28 for RCP 4.5 and 20/29/48 and 18/31/57 for RCP 8.5. From the results presented above, we can see that the range of increase in annual flow is more in

RCP 8.5 (5–57%) than in RCP 4.5 (10–30%), except in IF and MF of the cold-dry and in IF of the warm-dry conditions. The lower values of the predicted flows in RCP 8.5 are attributed to less precipitation in RCP 8.5 in these cases. Although the overall trend of the projected flow is found increasing, the individual scenarios show differences in the magnitude of changes in flows. Depending on the GCM used and the location of the studied catchments, the magnitude (in some cases even the direction) of changes in flow as the impact of CC are reported differently in previous studies [38,40,68–70]. For example, in the study by [69] using the SWAT model, the authors found that the annual runoff of the Yinma River Basin (China) in the future (2021–2050) would increase by 88% for RCP 4.5 and by 48% for RCP 8.5 in comparison to the baseline period (1981–2010). On the other hand, decreases of mean annual runoff are projected by the VIC model in all future time windows of 2010–2039, 2040–2069, and 2070–2099 in a similar study by [70] conducted in the Upper Yangtze River Basin of China (the decrease in mean annual flow was 7.84% under RCP 8.5 and 9.81% under RCP 4.5, in their case). Phi Hoang et al. [68] found that the ensemble flow due to CC shows increases in annual river flows between +5 and +16% in the Mekong river. Results from these studies highlight the need for the localized prediction of future flows for water resource management considering the uncertainties.

Except in May and June, the future flows predicted by all GCMs are likely to increase in all other months of the year for all time windows. This is similar to the results of the Koshi Basin in Nepal [40]. As in the case of annual flows, the range of predicted changes in future flows shows a high level of uncertainty depending on the choice of GCM. Such variation is quite high in RCP 8.5 (–50% to +200%). However, similar monthly variations in flow between −70% and +190% are found in Wagener et al. [71]. Similarly, maximum monthly flow increases of 143% and 99%, respectively, for RCP 4.5 and 8.5 were reported in [69]. On the other hand, [72] found that the mean monthly river flow varies from −16% to +55%, with the greatest decreases in July and August and the greatest increases in May and June.

Our results predict a higher increase in high flows than that of low flows. The average increases (predicted by all GCMs for all time windows) of high flow ($Q_{10}$) are 23 and 26%, and those of low flow ($Q_{90}$) are 14 and 17% with respect to the baseline for RCP 4.5 and RCP 8.5, respectively. This shows that the negative impacts of CC can be expected in both the high flows (increasing) and low flows (decreasing). Similar results are reported for the Ljubljanica River of Slovenia in all three investigated future time windows, i.e., 2011–2040, 2041–2070, and 2071–2100 under RCP 4.5 [73]. The highest change in one-day-maximum flood is expected in the warm-wet climatic condition for the case of RCP 4.5 in all time windows. w However, for RCP 8.5, the warm-dry condition in IF and MF predicts higher values of flood, while the cold-wet does so in the FF.

The expected rise in temperature will most likely lead to increase in water demand: for example, ET. This will further stress the water availability of the basin. On the other hand, increased snow melt during the dry season and thus addition to the current water availability might be beneficial to some users, such as hydropower projects. Moreover, the projected shift in precipitation patterns will most likely impact floods and droughts by altering their timing and magnitude. These consequences are further exacerbated by the various uncertainties in the method of analysis and results. Nevertheless, the increase in predicted floods by all the GCMs in our study show that the flood disposal structure should be designed at a higher capacity than the one designed based on baseline flood values to achieve climate resilience. Additionally, the projected decrease in the future low-flows due to CC strongly indicates the need for storage over run-of-river projects for optimal water use planning. Overall, it is seen that the impacts of CC are essential for the design of hydraulic structures, flood and drought management, and overall water resource planning and development of the basin.

## 5. Conclusions

The impact of CC on future water availability in the BRB was analysed by using a well-calibrated and validated SWAT hydrological model. Climate data were projected by four GCMs representing cold-dry, warm-dry, cold-wet, and warm-wet conditions for two emission scenarios, i.e., RCPs 4.5 and 8.5, adopting an envelope method. Most of the selected GCMs predict an increase in annual precipitation and temperature for RCP 4.5. In the case of RCP 8.5, most of the GCMs predict higher annual precipitation and temperature compared with the baseline condition, while some project a decrease in annual precipitation. The projected precipitation by almost all GCMs for all time windows (IF, MF, and FF) is most likely to increase in both RCPs. Both the maximum and minimum temperature projections by all the selected GCMs (in all climatic conditions) are found increasing while moving from IF to FF in both emission scenarios.

This study concludes that the increasing temperature and variation in precipitation patterns in the BRB resulting from CC will impact the water resource availability in the future. The monsoon flow is expected to increase significantly for all time windows in the case of both RCPs. While the variation in the monthly flows from the baseline values of RCP 8.5 is projected to be higher than that of RCP 4.5, the rate of increase is found to be more in the post-monsoon season. The greatest magnitude of the projected monthly maximum flow in all time windows for all climatic conditions except FF of the cold–wet condition is found to occur in August, similar to the baseline condition. The long-term average annual flow predicted by all GCMs for all time windows in RCP 4.5 and 8.5 is projected to continuously increase from IF to FF. The relative change in the mean monthly flow under RCP 4.5 and 8.5 is projected to increase for IF, MF, and FF. Ensembled flows are expected to be higher in RCP 8.5 than in RCP 4.5.

The fractional differences ($Q_{10}$:$Q_{90}$) for the projected flow were found to be progressively increasing with RCPs and over time. Additionally, the number of days exceeding the 10th percentile ($Q_{10}$, high flow) and not exceeding the 90th percentile ($Q_{90}$, low flow) are predicted to be more by all the GCMs. Likewise, the one-day-maximum floods of different return periods are projected to be higher than those of the baseline floods for all climatic conditions and for all time windows in both RCPs. Similarly, the one-day-minimum flows for different return periods are most likely to be lower than the base case. However, the predicted values are different in magnitude and direction depending on the selected GCMs.

The increase in future predicted floods implies the designing of flood disposal structures at a higher capacity than those designed based on historical data. Furthermore, the decrease in projected firm flows in the future suggests that storage-type water resource projects are preferred over run-of-river projects for optimal water use planning from the perspective of climate resilience.

**Supplementary Materials:** The following are available online at https://www.mdpi.com/article/10.3390/w13111548/s1. Table S1. Detail of scores for the changes in precipitation and temperature indices of selected GCMs in each corner before intermediate selection step (Step 2); Table S2. Selected GCMs in intermediate selection step (Step 3) using Taylor score; Table S3. The long-term monthly flow of three-time windows projected by four GCMs representing four climatic conditions of RCP 4.5 emission scenario; Table S4. The long-term monthly flow of three-time windows projected by four GCMs representing four climatic conditions of RCP 8.5 emission scenario

**Author Contributions:** Conceptualization, S.M., D.A. and L.P.D.; methodology, S.M., D.A., L.P.D., U.B. and D.S.; software, validation, formal analysis, investigation, resources, data curation, writing—original draft preparation, S.M., D.A., L.P.D., U.B. and D.S.; Writing—Review and editing, S.M., D.A. and L.P.D.; supervision, D.A. and L.P.D. All authors have read and agreed to the published version of the manuscript.

**Funding:** This research was partially funded by Tribhuvan University, Nepal.

**Institutional Review Board Statement:** Not applicable.

**Informed Consent Statement:** Not applicable.

**Data Availability Statement:** The datasets used in this study can be accessed freely.

**Acknowledgments:** The authors would like to thank the Department of Hydrology and Meteorology, Government of Nepal, for providing hydroclimatic data. Budhigandaki Hydroelectric Project Development Committee is also acknowledged for providing hydrological data. The authors would like to thank three anonymous reviewers for constructive comments which helps to improve our manuscript substantially.

**Conflicts of Interest:** The authors declare no conflict of interest.

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
