# Peer review of "Application of SWAT in Hydrological Simulation of Complex Mountainous River Basin (Part II: Climate Change Impact Assessment)"

_water, doi:10.3390/w13111548_

Round 1

Reviewer 1 Report

The second part of SWAT hydrological model in Mountainous River Basin of central Nepal deals with climate change impact assessment. It is an interesting article. The concerns for this article are:

In Introduction section, please also include the present climate change issues in the study area or Himalayan region.

Line 42……Delete (CC) as you have already used in Line 39.

Line 126……as described in “a study of…..” Give author name

Line 139……10th and 90th……Use superscript for “th”

Please try to improve the quality of the figures. It would be more scientific if you show the legends in the figure itself rather writing in Figure title. Show the lines (black colour) along with tick mark of x-axis and y-axis for all the graphs applicable, especially in Figure 5 and 7.

In case of Figure 4, Percentile 1-99 should be written in the last row to avoid any confusion. This should be just like the pattern followed in Figure 5.

In Figure 6, Time is Year while in In Figure 7, Time is Month. So please mention these in brackets.

Line 282…..Marahatta et al., 2021b……Please follow the MDP style for citation.

Table 5…..Use superscript m3/s in Baseline Flow Unit

Line 428……a similar study by…..Write the author name

Please avoid unnecessary use of bold in case of words throughout the manuscript. Check the Journal’s guidelines.

Please discuss the model validation.

The conclusive statement related to Immediate Future, Mid Future, and Far Future projections are missing.

Please summarize the future monthly, seasonal, and annual flows in conclusion to get a clear picture.

Check CO2 in Line 509 and 538. Use subscript for 2.

In many references, the year is not bold, e.g., 7-14, 18, etc.

Reference 19 is incomplete…… Part I : Formulation and…..?

Line 561…….Delete 00 Volume no. You can give the DOI.

Author Response

Reviewer's comments on Paper II

Reviewer 1: The second part of SWAT hydrological model in Mountainous River Basin of central Nepal deals with climate change impact assessment. It is an interesting article. The concerns for this article are:

Thank you for your appreciation on the article. We would like to respond to the comments as follows:

SN

Comments

Response

1

In Introduction section, please also include the present climate change issues in the study area or Himalayan region

In the Introduction section, the following sentences have been added in L87-93 of the revised manuscript:

Several studies have been conducted to assess the water availability and impacts of CC in the Hindu Kush Himalayan region [30–33] that includes Budhigandaki River Basin (BRB) [34–36]. Results of such studies vary considerably across the spatial and temporal scales and thus a generic conclusion on the impact of CC in water availability cannot be reached deterministically. These studies suggest the SWAT model can be a useful tool to assess the flow along with water balance and impacts of CC on them.

2

Line 42……Delete (CC) as you have already used in Line 39

Comments incorporated.

3

Line 126……as described in “a study of…..” Give author name

We have revised L131-132 in the revised manuscript as:

This study used the advanced envelop-based climate selection method to assess the projected future climates as described in Lutz et al. [28].

4

Line 139……10th and 90th……Use superscript for “th

It is corrected in the revised document as:

10th and 90th

5

Please try to improve the quality of the figures. It would be more scientific if you show the legends in the figure itself rather writing in Figure title. Show the lines (black colour) along with tick mark of x-axis and y-axis for all the graphs applicable, especially in Figure 5 and 7

Most of the suggested revisions have been incorporated in the final manuscript. However, the legends are intentionally omitted from some figures and explained in the caption to avoid congestion in the figure. 

6

In case of Figure 4, Percentile 1-99 should be written in the last row to avoid any confusion. This should be just like the pattern followed in Figure 5

Comments incorporated.

7

In Figure 6, Time is Year while in In Figure 7, Time is Month. So please mention these in brackets

Comments incorporated; time (Year and Month ) changed.

8

Line 282…..Marahatta et al., 2021b……Please follow the MDP style for citation

Comments incorporated.

9

Table 5…..Use superscript m3/s in Baseline Flow Unit

Comments incorporated

10

Line 428……a similar study by…..Write the author name

Comments incorporated in L434 in the revised manuscript.

11

Please avoid unnecessary use of bold in case of words throughout the manuscript. Check the Journal’s guidelines

Comments incorporated throughout the manuscript.

12

Please discuss the model validation

A well calibrated and validated SWAT model has been used for hydrological modeling of the study basin. The details of the model calibration and validation are given in Part-I of this paper. We have simply used the outputs of the CC models. 

13

The conclusive statement related to Immediate Future, Mid Future, and Far Future projections are missing

The following sentences have been added to the appropriate location of the Conclusion section:

  1. [L467-468] Projected precipitation by almost all GCMs for all time windows (IF, MF and FF) is most likely to increase in both RCPs.
  2. [L468-470] Both the maximum and minimum temperature projection by all the selected GCMs (in all climatic conditions) are found increasing while moving from IF to FF in both emission scenarios.
  3. [L478-479] The long-term average annual flow predicted by all GCMs for all time windows in RCP 4.5 and 8.5 are projected to continuously increase from IF to FF.
  4. [L479-481] The relative change in mean monthly flow under RCP4.5 and 8.5 is projected to increase for IF, MF and FF. Ensembled flows are expected to be higher in RCP 8.5 than in RCP 4.5.

14

Please summarize the future monthly, seasonal, and annual flows in conclusion to get a clear picture

 Please refer to response of #13.

15

Check CO2 in Line 509 and 538. Use subscript for 2

Comments incorporated

16

In many references, the year is not bold, e.g., 7-14, 18, etc

Comments incorporated

17

Reference 19 is incomplete…… Part I : Formulation and ..?

Comments incorporated

18

Line 561…….Delete 00 Volume no. You can give the DOI

Part I of this series paper, still not get DoI

Marahatta, S.; Aryal, D.; Devkota, L.P. Application of SWAT in Complex Mountainous River Basin (Part I :Model Development). Water 2021, doi:……

Reviewer 2 Report

The paper presents a climate change study using a hydrologic model in a mountainous area across Nepal and China. The paper is relatively well written. I have two major comments: 1. the paper is still verbose; 2. the discussion has been focused on the discharge and it is predicted to increase in nearly all climate change scenarios. Given that, the water budget change will be interesting. Authors can explain how each component (ET, snowmelt, baseflow etc.) in the water budget change under different scenarios. This will give the readers a comprehensive picture of the hydrologic responses due to climate changes.

Author Response

Reviewer's comments on Paper II

Reviewer 2: The paper presents a climate change study using a hydrologic model in a mountainous area across Nepal and China. The paper is relatively well written. I have two major comments: 1. the paper is still verbose; 2. the discussion has been focused on the discharge and it is predicted to increase in nearly all climate change scenarios. Given that, the water budget change will be interesting. Authors can explain how each component (ET, snowmelt, baseflow etc.) in the water budget change under different scenarios. This will give the readers a comprehensive picture of the hydrologic responses due to climate changes.

Thank you for your appreciation of our work. The main focus of this paper (Part-II) is to assess the impact of CC on future flows of the study basin. Therefore, our analysis, results and discussion are aligned with this objective and does not intentionally include the analysis of water balance. It is also to be noted that considering water balance and its components in addition to the different aspects of flow incorporating the results of the different scenarios will be too lengthy to accommodate in a single paper. Therefore, as a continuation of this research, we are planning to write one additional paper focusing on the water balance and its hydrological components considering the four climatic conditions and two RCPs. As the reviewer has indicated, we also strongly believe that the comprehensive contents of this planned paper will be interesting to the readers.

Hence, no further changes are made to the current paper in this regard.

Reviewer 3 Report

Considering the problems in Part I of this series papers, the authors are suggested to improve the modeling in Part I, and then reanalyze results in Part II for further review.

Author Response

Reviewer's comments on Paper II

Reviewer 3: Considering the problems in Part I of this series papers, the authors are suggested to improve the modeling in Part I, and then reanalyze results in Part II for further review.

Thank you for your comment. All the valuable comments and suggestions from the reviewers have been duly incorporated in the revised paper (Part-I). The SWAT model performance was reconfirmed and the results were found to be satisfactory. As suggested by the reviewer, the CC model runs and the consequent results were also revisited. We did not find any discrepancies in the results compared to those presented in this paper (Part-II). Hence, no modification was felt necessary and the current version of this paper (Part-II) is retained. 

Round 2

Reviewer 1 Report

The author has incorporated all the changes as suggested by me. I recommend accepting the manuscript in the present form.

Author Response

Dear Reviewer

Thank you for your appreciation and accepting the manuscript of the article. We have checked the minor English language and grammar as per your suggestion.

Reviewer 3 Report

This paper assessed the impact of climate change on runoff in a mountainous river basin in Nepal using calibrated SWAT model. The results can be used as a reference in water management, and the manuscript can be considered for publication after moderate revisions.

(1) More comparisons of the present results with available studies for similar rivers should be presented in the discussion section.

(2) Do not use too much abbreviations in the manuscript.

Author Response

Dear Reviewer,

Thank you for your appreciation on the article. We have checked the minor English language and grammar We would like to respond to the comments as follows;

We have added two paragraphs in the discussion as per the comments (line no. 412-426) and (line no. 473-486). Besides, we have reduced some abbreviation.
